# Molecular Evolution of Histone Methylation Modification Families in the Plant Kingdom and Their Genome-Wide Analysis in Barley

**DOI:** 10.3390/ijms24098043

**Published:** 2023-04-28

**Authors:** Bingzhuang An, Haiya Cai, Bo Li, Shuo Zhang, Yonggang He, Rong Wang, Chunhai Jiao, Ying Guo, Le Xu, Yanhao Xu

**Affiliations:** 1Hubei Key Laboratory of Food Crop Germplasm and Genetic Improvement, Key Laboratory of Ministry of Agriculture and Rural Affairs for Crop Molecular Breeding, Food Crops Institute, Hubei Academy of Agricultural Sciences, Wuhan 430064, China; 2College of Agriculture, Yangtze University, Jingzhou 434025, China; 3Scientific Observation and Experiment Station for Crop Gene Resources and Germplasm Enhancement in Hubei, Ministry of Agriculture and Rural Affairs, Hubei Academy of Agricultural Sciences, Wuhan 430064, China

**Keywords:** evolution, histone methylation, HMT, HDM, genome-wide analysis, barley

## Abstract

In this study, based on the OneKP database and through comparative genetic analysis, we found that HMT and HDM may originate from Chromista and are highly conserved in green plants, and that during the evolution from algae to land plants, histone methylation modifications gradually became complex and diverse, which is more conducive to the adaptation of plants to complex and variable environments. We also characterized the number of members, genetic similarity, and phylogeny of HMT and HDM families in barley using the barley pangenome and the Tibetan Lasa Goumang genome. The results showed that HMT and HDM were highly conserved in the domestication of barley, but there were some differences in the Lasa Goumang SDG subfamily. Expression analysis showed that *HvHMTs* and *HvHDMs* were highly expressed in specific tissues and had complex expression patterns under multiple stress treatments. In summary, the amplification and variation of HMT and HDM facilitate plant adaptation to complex terrestrial environments, while they are highly conserved in barley and play an important role in barley growth and development with abiotic stresses. In brief, our findings provide a novel perspective on the origin and evolutionary history of plant *HvHMTs* and *HvHDMs*, and lay a foundation for further investigation of their functions in barley.

## 1. Introduction

Histone methylation and demethylation modifications are the main epigenetic mechanisms, often referred to as the “second genetic code”, and play important roles in regulating transcription, genome integrity and epigenetics [1,2,3]. Histone methylation is dynamically and reversibly regulated by histone methyltransferases (HMTs) and histone demethylases (HDMs). HMT contains SDG (SET domain group) and PRMT (protein arginine methyltransferase) subfamilies. The HDM contains the HDMA (SWIRM and C-terminal domain) and JMJ (JmjC domain-containing proteins) subfamilies [3]. Histone lysine modification is catalysed by *SDGs*, which have been classified into seven classes in previous studies in plants such as *Arabidopsis* [4], *Brassica napus* [5] and *Dendrobium catenatum* [6]. Class I members, E (Z) homologues, can transfer methyl groups to H3K27; class II members, ASH1 homologues, are responsible for H3K4/H3K36 methylation; class III members, Trx homologues and related proteins are responsible for H3K4me1/3; class IV members, plant-specific proteins that contain SET and PHD domains, are responsible for monomethylation at H3K27; class V, SU (var) homologues are responsible for H3K9 methylation; and class VI and VII genes include disrupted SET domains (S-ET) or poorly functionally defined SET-related domains [6]. Arginine methylation is catalysed by a family of enzymes known as protein arginine methyltransferases (*PRMTs*). *PRMTs* are classified as type I or type II, depending on the position of the methyl group on the guanidine of the methylated arginine [7]. Seven type I arginine methyltransferases (*PRMT1a*, *PRMT1b*, *PRMT3*, *PRMT4a*, *PRMT4b*, *PRMT6* and *PRMT10*) and one type II enzyme (*PRMT5*) have been found in Arabidopsis [2]. On the other hand, histone methylation can be removed by the action of histone demethylases (HDMs) [8]. HDMA is a flavin adenine dinucleotide (FAD)-dependent enzyme that can only remove single/double lysine residue methylation. However, JMJ subfamily proteins that use Fe (II) and -ketoglutarate (KG) as cofactors will catalyse the removal of mono/di/tri-lysine residue methylation [1]. The JMJ subfamily can be divided into five categories according to sequence similarity and catalytic specificity in plants [1]. KDM5/JARID1 group proteins have been found to catalyse the demethylation of H3K4me1/2/3, KDM4/J*HDM*3 family proteins have been shown to actively demethylate histone H3K9me2/3, H3K36me2/3 and KDM3A/JHDM2 can demethylate H3K9me2 and H3K9me1 but not H3K9me3, the JMJD6 group can demethylate H3R2me2 and H4R3me2, and the JmjC domain-only subfamily can remove the methylation of H3K27me3 [3].

A combination of epigenetics and environmental stress dynamics has led scientists to investigate the impact of epigenetics on evolution in plants [9]. Theoretical models and simulations predict that epigenetic variation has the potential to affect the rate and outcomes of adaptation [10,11,12]. The adjustment of gene expression is a key mechanism used by plants during growth and developmental processes. Because plants are immobile organisms, control of gene expression becomes more essential when plants are subjected to inescapable environmental stressors [13,14]. Histone methylation can promote plant adaptation to stressful environments by regulating the expression of stress-responsive genes [15,16]. ARABIDOPSIS TRITHORAX4 (*ATX4*) and *ATX5* play essential roles in the drought stress response [17]. Atx4 and atx5 single loss-of-function mutants showed drought stress-tolerant and ABA-hypersensitive phenotypes during seed germination and seedling development [17]. Downregulation of stress-responsive genes upon dehydration treatment is accompanied by an obvious decrease in H3K4me and H3K4me3, while H3K4me2 levels are only slightly reduced, indicating that *HDMs* might modulate the expression of dehydration stress-responsive genes [18]. During salt stress, *SKB1* mediates the salt response by altering the methylation status of H4R3sme2 of stress-responsive genes [19]. Partial histone methylation modifications have been shown to be missing in some algae, suggesting that plants may have acquired more diverse histone methylation modifications through evolution, and that this evolution may have facilitated plant adaptation to adversity [20,21]. However, studies have mainly focused on angiosperms, and nothing is known about the origin of HMT and HDM in the plant kingdom and the evolutionary relationship between species. Further studies on the evolutionary history of HMT and HDM will help to better understand their important roles in plant evolution.

Barley (*Hordeum vulgare* L.) is among the world’s earliest domesticated crop species, being cultivated in the Nile River Valley of Egypt at least 17,000 years ago [22]. It is the world’s fourth largest cereal crop and has many uses, such as feeding, food and beer [23]. Grain yield is the most important driver of grower profitability and is thus a key breeding target for genetic improvement. Yield is, however, a challenging breeding target as it is controlled by a large number of genes with varying effects, and strongly influenced by genotype and by environmental interactions (complex traits) [24]. Research has indicated that histone methylation can regulate gene expression, thereby modulating the growth and development of barley and its adaptation to stress. High chromatin H3K27me3 levels before vernalization inhibited the expression of the *HvVRN1* gene that induced vernalization in barley (Hv-*Hordeum vulgare* L.), and after vernalization, chromatin H3K4me3 levels increased, while H3K27me3 levels decreased, promoting its expression [25]. Under salt stress, salt-tolerant varieties of rye were significantly enriched in H3K4me3 and H3K27me3 compared to salt-sensitive varieties, and the significant enrichment of both modifications was associated with oxidative processes [26]. However, the characterization of barley HMT and HDM families has not yet been reported. Furthermore, barley has a large and complex genome, similar to other *Triticae* crops such as wheat, making genome-based selection difficult. In recent years, the publication of the barley pangenome and the Tibetan naked barley (Lasa Goumang) genome has greatly facilitated the genetic investigation of barley [27,28]. Limited research has been conducted on the comprehensive characterization of gene families across various germplasms. The identification and analysis of HMT and HDM in multiple germplasms are particularly valuable for comprehending their significant roles in the evolutionary process of barley adaptation.

In this study, we analysed the origin of HMT and HDM in the plant kingdom and the evolutionary relationship between different species. The HMT and HDM families in barley were characterized through the barley pangenome and the Tibetan barley variety (Lasa Goumang), and their expression profiles in different organs and under various abiotic stresses were also analysed. This provides a basis for future exploration of the regulatory role of HMT and HDM in barley growth, development and stress response.

## 2. Results

### 2.1. Origin and Evolution of HMT and HDM in Green Plants

To explore the origin of HMT and HDM in the plant kingdom, we performed a comparative genetic analysis among green plants [29]. The amino acid sequences of 55 *HMTs* and 29 *HDMs* from Arabidopsis were set as seed sequences to search the OneKP Phytozome genome database. Phylogenetics showed that HMT and HDM are conserved in land plants and algae, and that both may have evolved from Chromista (such as *Petalonia fascia*, *Scytosiphon dotyo*, *Colpomenia sinuosa*, etc.) (Appendix A). Phylogenetic topology clearly separates algae from land plants. To further explore the evolutionary relationship between HMT and HDM between algae and land plants, a total of 3846 homologous genes were identified from 35 species, including monocots, dicots, gymnosperms, ferns, lycophytes, mosses, green algae, red algae, and brown algae (Figure 1a). Dramatic changes in the environment accompanied the evolution from algae to land plants; asteroids hit the Earth several times, and a particularly severe impact preceded the emergence of land plants, with continual increases in atmospheric oxygen, fluctuations in carbon dioxide levels, and continually rising light intensity (Figure 1a) [30]. The number of HMT and HDM family members among species was significantly higher in land plants than in algae, including the gene copy number of SDG, JMJ, PRMT (there were slightly fewer ferns and gymnosperms) and HDMA subfamilies (Figure 1a). Therefore, a comparative genetic analysis grouped subfamilies according to Arabidopsis classification based on the multiple protein sequence ratios of 35 different species, with all Arabidopsis HMT and HDM gene families as references (Figure 1b). We found that some subgroup members of the SDG and JMJ subfamilies were missing in algae. Among the SDG subfamily, class I and class IV share H3K27 methylation modifications, but their homologues were not found in brown algae. In addition, although class I and class IV share H3K27 methylation modification, class IV appeared late in the evolution of species, and its homologues were not found throughout algae until land plants. Similarly, class ii and class III share H3K4 methylation modification, but class III appears later than class II. This suggests that as species evolved, the same histone methylation modification changed from being regulated by a single grouping to being regulated by multiple groupings, which contributed to a more complex modification mechanism of histone methylation. Class V (modified H3K9 methylation) is the largest subfamily, including SUVH and SUVR homologues. SUVR is conserved in algae, but SUVH is only found in *Chlamydomonas* (green algae). In the JMJ subfamily, KDM3 (modified H3K9me1/2) was missing, and its homologues were also found only in *Chlamydomonas*. Hence, as algae evolved to land plants, family expansion led to the emergence of several new groupings and homologues.

### 2.2. Number of HMTs and HDMs in Barley

A total of 1671 homologous genes were identified from 21 materials of the barley pangenome and Lasa Goumang (Table 1; Appendix A). In *HvHMTs*, there were a total of 7 *HvPRMTs* in all germplasms, with between 49 and 53 *HvSDGs* in wild and cultivated barley (except Lasa Goumang), while in Lasa Goumang, there were significantly fewer *HvSDGs* than in other germplasms (only 42). Regarding *HvHDMs*, there were 4 *HvHDMAs* in all germplasms, with between 17 and 19 *HvJMJs* members in all germplasms.

### 2.3. Variation in HMT and HDM Family Members in Barley

To investigate the variation among different germplasms, *HvHMTs* and *HvHDMs* in Morex were used as a reference to observe the sequence homology of homologous genes among different germplasms (Figure 2). The homology of most genes was above 99%, including the homology found in the HvHDMA and HvPRMT subfamilies. However, the individual gene sequences of HvJMJ and HvSDG had clear variations; for example, the homologous gene sequence identity of *HORVU.MOREX.r3.1HG0093350.1* (SDG) in B1K-04-12 was 26.0%, and that of *HORVU.MOREX.r3.5HG0474610.1* (SDG) in HOR_13821 was 69.1%, whereas the homologous gene sequence homology of *HORVU.MOREX.r3.4HG0411310.1* (JMJ) in Lasa Goumang was 44.1%. These results indicate that homologous genes among different barley germplasms are highly homologous, but few genes have undergone gene sequence variation.

### 2.4. Phylogenetic, Conserved Domain and Motif Analysis of HMT and HDM in Barley

To investigate the structural characteristics and phylogenetics among the *HvHMTs* and *HvHDMs*, the 217 HMT and 89 HDM protein sequences from wild barley (B1K-04-12), cultivated barley (Morex), Tibetan barley (Lasa Goumang) and Arabidopsis were used (Figure 3 and Appendix A). The HvHMT family is composed of two subfamilies, HvSDG and HvPRMT. All *HvSDGs* are classified into seven groups, each of which contains group-specific conserved domains in addition to the SET domain. Class I contains CXC and SET domains; class II contains zf-CW, Pre-SET and AWS domains; class III contains zf-HC5HC2H_2, PWWP and PHD domains; class IV contains the Zf_RING and PHD domains; the class V subgroup I contains the Pre-SET and SAD_SRA domains, and subgroup II contains the Pre-SET and WIYLD domains; and classes VI and VII contain the Rubis-subs-bind and zf-MY ND domains. *HvPRMTs* were highly conserved, including PrmA, Methyltransf_25 and PRMT5 domains, and they were divided into type I and type II. Phylogenetic analysis revealed that all At*PRMTs* have homologous genes in barley, except *AtPRMT7*. The HvJMJ subfamily in HvHDM was divided into five groups, and KDM4 and KDM5 could be further divided into two subgroups. KDM3 features the zf-4CXXC_R1 domain and the JmjC domain. The *HvJMJs* of KDM4 all contain JmjC, JmjN and zf-C2H2 domains, and the *HvJMJs* of the KDM4 class falls into two main subclasses. Subgroup I is characterized by the zf-H2C2_2 domain, while subgroup II contains the zf-C5HC2 domain. The *HvJMJs* of the KDM5 class all contain JmjN, zf-C5HC2 and Jmjc domains. Additionally, subgroup I contains FYRN and FYRC domains, while subgroup II contains 1 ARID, 2 PHD and 2 PLU-1 domains. Jmjc-only contains only one Cupin_8 domain; JMJD6 contains cupin_Rmlc-like, F-box-like and APH domains. All genes of HvHDMA were highly conserved, containing only one N-terminal Amino_oxidase and one C-terminal SWIRM domain.

The above results indicate that HMT and HDM are highly conserved in barley. However, domain observation among wild barley, cultivated barley, and Lasa Goumang revealed that domestication also led to the emergence of some new structural domains and the loss of some structures of a few genes in the HvSDG and HvJMJ subfamilies, with more significant losses in the SDG subclade of Lasa Goumang. For example, Morex and Lasa Goumang contain the SPICE domain (JMJ subfamily), which is missing in wild barley B1K-04-12; Morex lacks the zf-TRM13_CCCH and zf-C2H2_3rep domains contained in Lasa Goumang and wild barley (SDG subfamily); and Lasa Goumang contains germplasm-specific Med15, tRNA_deacylase and SAWADEE domains (SDG subfamily). In addition, we explored the source of deletion of SDG family members in Lasa Goumang and found that the missing genes in Lasa Goumang were mainly from class II, and AWS and zf-CW domains unique to class II were missing (Appendix A, Figure 3).

To better understand the conservation and diversification of *HvHMTs* and *HvHDMs*, the putative motifs of all HvHMT and HvHDM proteins were predicted by MEME motif analysis (Appendix A). HvHMT and HvHDM proteins in the same group had similar motifs, which aligned with the results of the phylogenetic analysis. Additionally, there were a few differences in motifs between the HvSDG and HvJMJ subfamilies among germplasms, and they were mainly concentrated in HvSDG subfamily genes of Lasa Goumang. In general, family members within the same group had similar functions.

### 2.5. Chromosomal Location Analysis of HvHMTs and HvHDMs

We performed a chromosomal localization analysis of histone methylation gene families in three different barley cultivars (Morex, B1K-04-12, OUN333) (Appendix A). The results showed highly similar distribution of homologous genes across the different cultivars. All genes were located on barley chromosomes 1 to 7. Chromosomes 3 and 5 contained the largest number of genes, while chromosome 4 had the fewest. SET subfamily genes were distributed on all chromosomes, PRMT subfamily genes were located on chromosomes 1, 2, 5, 6, and 7, JMJ subfamily genes were located on all chromosomes except chromosome 2, and HDMA subfamily genes were located on chromosomes 2, 6, and 7. The gene distribution of HMT and HDM subfamilies among different germplasms was highly consistent, which indicated that HMT and HDM subfamilies were highly conserved in barley. In the barley genome, genes with different histone methylation modifications are distributed on different chromosomes, which may reflect their functional differences in different biological processes.

### 2.6. Analysis of Cis-Acting Elements of HvHMTs and HvHDMs

Plant growth and development are regulated by different *cis*-regulatory elements in genes. Various *cis*-acting elements, containing hormone-sensitive, light-responsive, stress-responsive, tissue-specific and other elements were mined and analyzed from the 2000-bp upstream regulatory regions of *HvHMTs* and *HvHDMs* genes using the PlantCARE web site (Figure 4). The most abundant *cis*-acting element was the light-responsive element; tissue-specific elements, including in the endosperm, meristems, roots, seeds and palisade mesophyll cells, were found in the putative promoters of HvHMT and HvHDM genes. Hormone-responsive elements, in response to auxin, ABA, GA, MeJA and SA, were also widely observed; multiple abiotic and biotic stress-related elements, including in response to drought, anaerobic, hypothermia, injury, defense and stress, were largely enriched. These results suggest that *HvHMTs* and *HvHDMs* might play important roles in plant growth and development, hormone signaling, and resistance to biotic and abiotic stresses.

### 2.7. HvHMTs and HvHDMs Expression Patterns in Different Organs

Analysis of gene expression patterns can provide directions and strategies for prediction of gene function [31]. The expression profiles of *HMTs* and *HDMs* at 16 developmental stages of barley were investigated using RNA-seq data (Figure 5). All *HvHMTs* and *HvHDMs* were expressed in at least one tissue or stage. The same genes, such as *HvJMJ*6, *Hv*SET15 and *Hv*SET38, were highly expressed in almost all tested tissues across diverse developmental stages; hence, they played an integral role in organ formation and development. Most genes were highly expressed in the developing inflorescence (INF1 and INF2), and histone methylation modification genes might play an important role in the development of barley inflorescence. In addition, expression profiling revealed that some genes were tissue- and stage specific. *HvSET48*, *HvSET6*, *HvSET36*, and *HvSET40* were only highly expressed in senescent leaves and etiolated seedlings, suggesting that they might have functions in regulating cell senescence and death. *HvSET2*, *HvSET46*, *HvJMJ*10, and *HvSET26* are highly expressed only in the early stage of grain development (CAR5), while *HvSET3*, *HvSET19*, and *HvSET31* are highly expressed in both stages of grain development (CAR5 and CAR15), indicating a specific role in grain development. *HvJMJ13*, *HvSET3*, *HvSET30* and *HvPRMT1*, *HvPRMT5* and *HvPRMT10* were also specifically expressed in the epidermis and roots of surrounding seedlings.

### 2.8. HvHMTs and HvHDMs Expression Profiling under Abiotic Stresses

To confirm whether *HvHMTs* and *HvHDMs* are involved in the stress response, comprehensive expression patterns were analyzed under different abiotic stresses, including low temperature, drought, salt, and waterlogging (Figure 6a). A total of 36 *HvHMTs* (30 *HvSDGs* and 6 *HvPRMTs*) and 18 *HvHDMs* (14 *HvJMJs* and 4 *HvHDMAs*) were differentially expressed under different stresses (Appendix A). Different response patterns were noted in a single subclade, and some were specifically stress-dependent. For example, of a total of seven members of the PRMT subfamily, six were significantly upregulated under low-temperature stress, whereas no members of the PRMT subfamily were differentially expressed under flooding stress. A total of four HDMA subfamily members were differentially expressed under low temperature, salt and flooding stresses, while no HDMA subfamily members were differentially expressed under drought stress. Among the differentially expressed genes, the expression patterns of certain genes differed under different stress treatments. For instance, *HvJMJ*4 expression was downregulated under drought stress but upregulated under low-temperature stress, and *HvSET20* expression was upregulated under low temperature stress but downregulated under salt stress. In addition, *HvHMTs* and *HvHDMs* were differentially expressed in different germplasms. Whereas *HvPRMT3*, *HvPRMT10*, *HvJMJ*4, and *HvSET22* were differentially expressed only in sensitive varieties under drought stress, under salt stress, *HvPRMT3*, *HvLDL1*, *HvJMJ1*, and *HvSET9* were differentially expressed only in tolerant varieties. These results suggest that *HvHMTs* and *HvHDMs* have complex expression patterns and are variety-dependent.

To eliminate intervarietal differences and to further validate the above results, we subjected the barley varieties of Efupi N0.1 to low temperature, drought, salt and flooding stress treatments. Furthermore, in addition to these stresses under natural conditions, we further explored the expression patterns of *HvHMTs* and *HvHDMs* under ionizing radiation (gamma) using different sensitive varieties (tolerant variety, 9117; sensitive variety, 9127). We selected 12 genes differentially expressed in the barley transcriptome under low temperature, drought, salt, and flooding for qRT–PCR, and the primers are listed in Appendix A (Figure 6b). For example, *HvLDL3* and *HvPRMT3* showed an expression pattern that peaked at 12 h under all four stresses except radiation. The upregulation of *HvPRMT3* was also similar, ranging from 17-fold to 20-fold under salt, drought, and flooding stresses. However, most of them showed specific expression patterns under different stress treatments; for example, *HvJMJ4* showed significant inhibition after 3 h of drought but was significantly induced at 3 h under low temperature, and *HvSET4* expression gradually increased at 0–12 h and significantly decreased after 24 h under low temperature and salt stress, while its expression continued to increase under drought and flooding stress and remained at a high level after 24 h. In addition, the expression of 13 genes under radiation was correlated with both material sensitivity and radiation dose, and *HvPRMT5* expression was significantly suppressed in the tolerant material (9117) at a dose of 50 Gy, while there was no difference in expression in the sensitive material (9127). In tolerant material (9117), *HvSET4* gene expression did not change significantly at 50 Gy, while expression was significantly repressed at 200 Gy. The above results indicate that some *HvHMTs* and *HvHDMs* had similar expression patterns, and that most genes had complex expression pattern variations. Therefore, differential expression of *HvHMTs* and *HvHDMs* under ionizing radiation further highlights that histone methylation modification is an epigenetic modification that widely responds to multiple stresses.

## 3. Discussion

Proof of similarity between eukaryotic and archaeal chromatin suggests that histone and chromatin structures evolved before archaea and eukaryotes diverged [32,33]. This also suggests that the initial function of nucleosomes and chromatin formation might have been the regulation of gene expression rather than the packaging of DNA. Histone methylation and acetylation are considered to be the most important and prevalent epigenetic markers associated with gene regulation [34]. The regulatory effects of histone methylation modifications can facilitate rapid adaptation of organisms to extreme fluctuations in environmental conditions [35]. In this study, we found that the evolution from algae to land plants was accompanied by great changes in the environment. At the same time, the colonization of terrestrial habitats by plants is accompanied by exposure to a number of abiotic stress factors, such as high irradiance, lack of mineral nutrients (bare rocks and no soils), varying temperature, and dehydration [36]. In this study, the number of *HMTs* (*PRMTs* and *SETs*) and *HDMs* (*JMJs* and *HDMAs*) was found to increase significantly during the aquatic-terrestrial transition from algae to land plants, while their sequence homology also changed significantly. In addition, all subfamilies and subgroups of HMT and HDM are commonly found in land plants, and only the KDM3 subgroup is missing in *Selaginella moellendorffii*. In algae, a large number of homologous genes of the SDG and JMJ subfamilies were missing. H3K27me3 was absent in brown algae, while H3K9me2/3 was detected only at very low levels, which is consistent with the results of the present study [37]. The homologue of E(z) (SDG class I) that controls H3K27 methylation was not detected in brown algae during this study. Furthermore, the SUVH (SDG) homologue was absent in brown algae, suggesting that low levels of H3K9me2/3 may be regulated by the SUVR homologue in class V (SDG). In whole algae, KDM3 (JMJ) and SUVH homologues are only present in *Chlamydomonas reinhardtii*, and the SUVH homologue SET3p has been reported, which functions in vitro as a specific H3K9 mono-methyltransferase [38]. Some subsequent subgroups, such as class III and class IV of the SDG subfamily, commodify H3K4 and H3K9 methylation with the previously existing class II and class I, which makes histone methylation modifications more complex and diverse. In summary, during the evolution of the species, HMT and HDM showed an amplified state, and along with the emergence of some new subgroups in the HMT and HDM subfamilies, histone methylation gradually became complex and diverse. The emergence of new resistance genes often takes hundreds of thousands of years, and in the face of abrupt and dramatic environmental changes, histone methylation modifications, which have become more complex and diverse with evolution, may be more rapid and effective in regulating the expression of existing resistance genes than in generating new resistance genes. Complex and diverse histone methylation modifications can promote better adaptation of plants to complex terrestrial environments and help them cope with sudden environmental stresses.

The domestication process of barley and its wide adaptability have long been the focus of researchers around the world. In the past decade, individual reference genomes have not been fully representative of species diversity due to the high degree of genomic variation. Sequencing of the barley pangenome and Lasa Goumang provides an excellent opportunity for this purpose [39]. On this basis, we identified HMT and HDM using the barley pangenome and the Lasa Goumang genome. Plant domestication is an evolutionary process whereby humans have used wild species to develop new and altered forms of plants with morphological or physiological traits that meet human needs [40]. The domestication process has resulted in reduced diversity at both the genome and local level. For example, more than half of genetic variation has been lost in cultivated soybean, 2–4% of maize genes have experienced artificial selection, and cultivated rice has also suffered genetic erosion [41,42,43,44]. In this study, we found that only a small number of genes in the SDG and JMJ subfamilies of wild barley and cultivated barley (except Lasa Goumang) have low homology, with few differences in the conserved domains. However, the number of family members is nearly identical, and most genes in farmed and wild germplasm are nearly identical. This suggests that the HMT and HDM families have an important role and remain highly conserved in the domestication process of barley. The PRMT, HDMA and JMJ subfamilies in Lasa Goumang were not significantly different from those in wild barley and cultivated barley (except Lasa Goumang). However, there were significantly fewer SDG subfamily members in Lasa Goumang, and they showed more variation than other germplasms, accompanied by the addition and loss of structural domains. Members of the PRMT, HDMA and JMJ subfamilies in Lasa Goumang were not significantly different from those in wild barley and cultivated barley, but the number of members of the SDG subfamilies in Lasa Goumang was significantly lower. At the same time, there is obvious addition and loss of domains, and the class II grouping of the SDG subfamily is missing. However, this phenomenon was not found for other barley varieties in the pangenome. Therefore, we speculate that this may be due to the loss of some SDG subfamily genes in Lasa Goumang during the artificial domestication process, or major mutations in some SDG subfamily genes, which made it impossible to identify them by Blastp.

Histone methylation can occur at various sites in histone proteins, primarily on lysine and arginine residues, and it can be governed by multiple positive and negative regulators, even at a single site, to either activate or repress transcription [45]. *Cis*-acting elements in promoter regions are always associated with their transcriptional control and function, and in this study, functional elements associated with tissue-specific stress responses were identified in the predicted promoters of *HvHMTs* and *HvHDMs*. The important role of histone methylation in regulating flowering, grain development, leaf senescence and root growth has been revealed in Arabidopsis, rice and Eucalyptus grandis, and the specific high expression of *HvHMTs* and *HvHDMs* in barley flowers, grains, senescing leaves and roots indicates their important role in barley growth and development [46,47,48,49]. Abiotic stresses, including heat, drought, cold, flooding, and salinity, negatively impact crop productivity. Histone methylation has been shown to represent key modulators in plant stress responses [50]. At low temperatures, COLD-REGULATED15A (COR15A) and GALACTINOL SYNTHASE3 (GOLS3) are activated by a decrease in the H3K27me3 modification level in the gene region, thereby helping plants to defend against coldness [51]. Under drought stress, drought response factors and genes in the ABA biosynthesis pathway are activated by H3K4me3 modification [52]. Salt stress generally results in the deposition of active histone markers such as H3K9K14Ac and H3K4me3, and reduces the deposition of H3K9me2 and H3K27me3 inhibitory histone markers on salt-tolerant genes [13]. Dynamic and reversible histone modifications occur in rice seedlings under submergence stress, and ADH1 and PDC1, which are submergence-inducible genes, respond to stress as a result of increased H3K4 methylation and H3 acetylation [53]. In this study, experimental data from RNA-seq and qRT–PCR showed that *HvHMTs* and *HvHDMs* exhibited significant transcriptional responses with complex expression patterns under abiotic stresses. Generally, histone H3K9 and H3K27 methylation is associated with transcriptional gene silencing, whereas H3K4 and H3K36 methylation is associated with gene activation [13]. This suggests that *HvHMTs* and *HvHDMs* may modify the histone methylation of key responsive genes under different stresses to improve plant stress resistance by regulating their transcriptional levels. The transcriptional responses of HvHMT and HvHDM to low temperature, drought, salt, waterlogging and ionizing radiation also indicate that HvHMT and HvHDM respond extensively to various stresses.

## 4. Materials and Methods

### 4.1. Evolutionary Analysis

The HMT and HDM proteins of Arabidopsis were downloaded from TAIR (https://www.arabidopsis.org/index.jsp (accessed on 15 April 2022)), which were used as query sequences, and candidate protein sequences orthologous to HMT and HDM proteins were searched using the One Thousand Plant Transcriptome (OneKP) database (https://db.cngb.org/blast/blast/blastp/ (accessed on 20 April 2022)) [54]. Multiple sequence alignments were conducted using MAFFT software V7 with the default parameters (https://mafft.cbrc.jp/alignment/server/ (accessed on 8 May 2022)). Phylogenetic trees were constructed using IQ-TREE v2.0.6, and support for each node was assessed by performing a bootstrap analysis with 1000 replicates [55]. The results were displayed using iTOL 6.0 visualization (https://itol.embl.de/ (accessed on 21 May 2022)) [56].

Genomes of 35 plant species from aquatic to land evolution were downloaded from the NCBI (https://www.ncbi.nlm.nih.gov (accessed on 3 June 2022)), Ensembl plant (https://plants.ensembl.org/index.html (accessed on 4 June 2022)), and Phytozome (https://phytozome-next.jgi.doe.gov/pz/portal.html (accessed on 5 June 2022)) databases. *HMTs* and *HDMs* of Arabidopsis were used as query sequences to predict the number of HMT and HDM proteins in different plant and algal species. Two methods were applied to identify the members of the HMT and HDM gene families in barley. In the first approach, the HMM files of HMT and HDM (HMT: SDG-PF00856, PRMT-PF05185; HDM: HDMA-PF04433, Amino_oxidase-PF05193, Jmjc-PF02373) were retrieved from the Pfam database (version 32.0; https://pfam.xfam.org/ (accessed on 23 July 2022)) and used as a query to search the barley proteins using HMMER software (version 3.3.2; https://hmmer.org/ (accessed on 28 July 2022)) with the default parameters. In the second approach, the Arabidopsis and rice HMT and HDM protein sequences were used to search against the barley protein sequences using Blast v2.9.0 software at an Evalue < 1 × 10^−5^ and coverage > 50% [57]. The HMT and HDM protein evolution time tree of the current phylogeny of plant species was constructed using the online tool TIMETREE (https://www.timetree.org/ (accessed on 15 August 2022)). Protein homology values with the highest similarity to Arabidopsis HMT and HDM proteins in 35 plant species were used to draw a protein similarity heatmap [57].

### 4.2. Biogenic Analysis of HMT and HDM in Barley

To identify HMT and HDM genes in *H. vulgare*, the barley pangenome and Lasa Goumang genome were downloaded from IPK (https://galaxy-web.ipk-gatersleben.de/libraries (accessed on 25 August 2022)) and NCBI (https://www.ncbi.nlm.nih.gov (accessed on 26 August 2022)). Blastp and HMMER search methods were used to identify HMT and HDM family homologous genes in barley. Overlapping genes identified by both approaches were considered the final members of the HMT and HDM gene families in barley.

To identify conserved protein domains, the Conserved Domains Database (CDD) of NCBI (https://www.ncbi.nlm.nih.gov/Structure/bwrpsb/bwrpsb.cgi (accessed on 2 September 2022)) and PfamScan (https://www.ebi.ac.uk/Tools/pfa/pfamscan/ (accessed on 5 September 2022)) with a threshold of Evalue < 0.01 was employed. Conserved motif identification within the HMT and HDM gene sequences was performed using the online tool MEME v5.4.1 (https://meme-suite.org/meme/tools/meme (accessed on 13 September 2022)) with a maximum number of 10. To identify promoter regulatory elements, the PlantCARE (https://bioinformatics.psb.ugent.be/webtools/plantcare/html/ (accessed on 15 September 2022)) online tool was used to annotate the 2 kb genomic region upstream of the initiation codon of the HMT and HDM genes.

The gene expression levels of 14 different growth stages and different tissue samples were obtained through the IPK Barley BLAST Server (https://galaxy-web.ipk-gatersleben.de/libraries (accessed on 23 September 2022)). RNA-seq data of abiotic stresses (salt, waterlogging, drought and cold) were downloaded from the NCBI Sequence Read Archive (http://www.ncbi.nlm.nih.gov/sra (accessed on 29 September 2022)) database (PRJNA578897, GSE144077, PRJEB40905 and PRJNA322603) to investigate the expression profiles of HMT and HDM genes in barley.

### 4.3. Plant Material, Stress Treatment, RNA Extraction, and qRT–PCR Analysis

Barley seeds (Efupi N0.1) were sterilized with 70% alcohol for one minute, rinsed three times with deionized water, sterilized with 2.5% sodium hypochlorite for 20 min, thoroughly rinsed with distilled water, and soaked for 30 min for sprouting. The soaked seeds were placed flat on a Petri dish with wet filter paper, at low density. Seeds germinated at 27 °C in the dark for two days. Then, plants were grown for 7 days under 16 h light: 8 h dark at 27 °C. For cold, drought and salt treatments, plants were incubated at 4 °C, with 20% PEG2000 and 200 mM NaCl. For the waterlogging treatment, the plants were moved to a growth box and covered with tap water to 1–2 cm above the soil surface level [58]. Seedlings at 0 h were considered the control, and seedling leaves from all treatments and the control were carefully harvested at 0, 3, 6, 12 and 24 h, immediately frozen in liquid nitrogen, and stored at −80 °C for subsequent analysis. For radiation treatment, the dry seeds of 9127 (radiation-sensitive) and 9117 (radiation-tolerant) were exposed to 7Li-ion beam gamma rays at doses of 50 Gy and 200 Gy, respectively.

Total RNA was isolated from barley leaves treated under various stress conditions using TRIzol reagent (TaKaRa, Gunma, Japan). A FastKing RT Kit (with gDNase) (TIANGEN, Beijing, China) was used to remove genomic DNA contamination and perform cDNA synthesis following the manufacturer’s instructions. Quantitative real-time PCR (qRT–PCR) was performed using SuperReal PreMix Color (SYBR Green) (TIANGEN, Beijing, China). The following PCR parameter conditions were set: denaturation at 95 °C for 2 min, 40 cycles of denaturation at 95 °C for 10 s, annealing at 60 °C for 40 s, and extension at 72 °C for 15 s. The relative expression level was calculated by the 2^−ΔΔCT^ method.

## 5. Conclusions

In the plant kingdom, HMT and HDM probably originated from Chromista and underwent significant amplification and mutation during evolution from algae to land plants, promoting a complex diversification of histone methylation modifications, which in turn facilitated plant adaptation to variable terrestrial environments. Meanwhile, identification and analysis of different barley germplasms revealed that although *HvHMTs* and *HvHDMs* were more conserved among different barley germplasms, some variation existed, with the most significant variation in Lasa Goumang. Both development and stress response showed distinct expression profiles throughout different tissues and under different abiotic stresses.

## Figures and Tables

**Figure 1 ijms-24-08043-f001:**
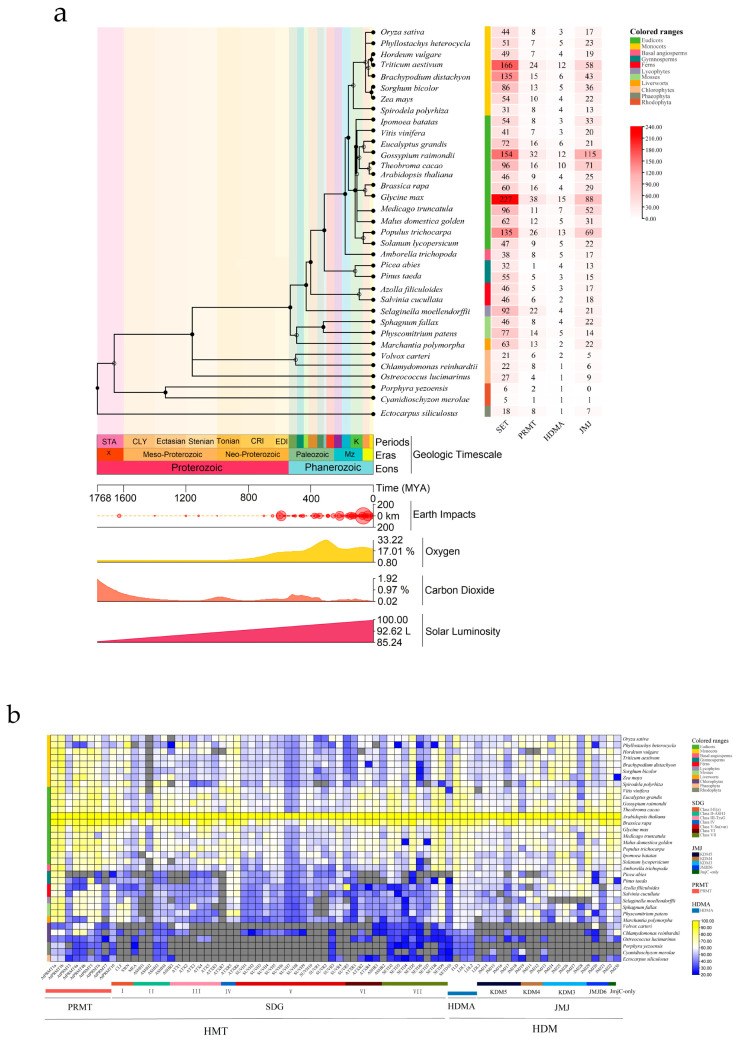
(**a**) Predicted number of HMT and HDM in different plant and algal species. Coloured squares indicate the protein predicted number from 0 (white) to 240 (red). (**b**) Similarity heatmap of HMT and HDM proteins in plant and algal species. Coloured squares indicate protein sequence similarity from zero (blue) to 100% (yellow). Gray squares indicate no proteins that satisfied the selection criteria.

**Figure 2 ijms-24-08043-f002:**
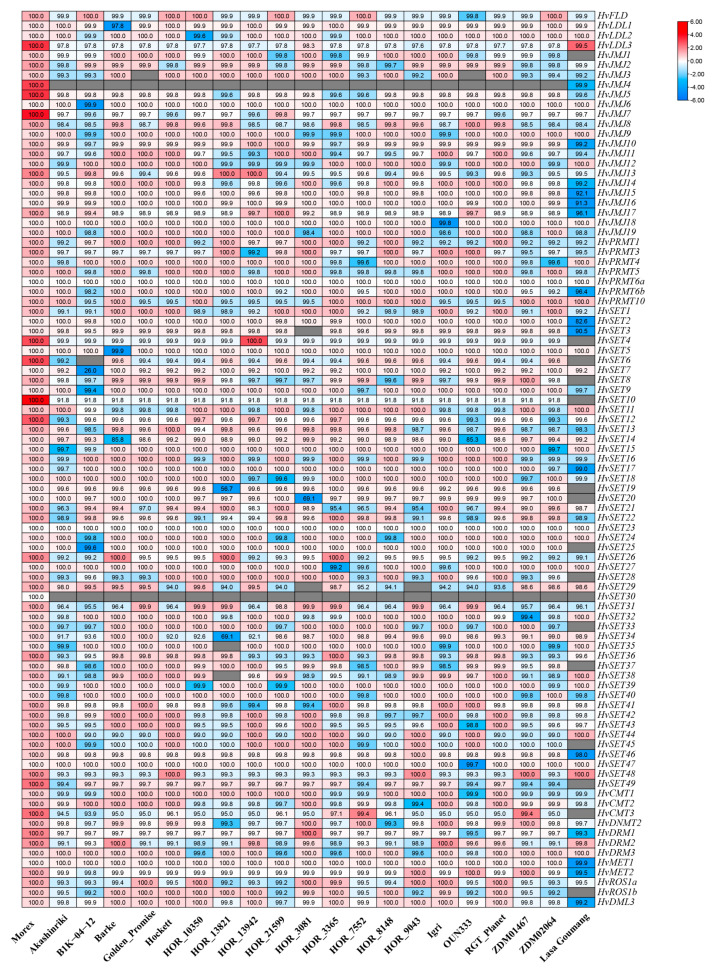
Sequence homology of homologous genes among different germplasms. Coloured squares indicate protein sequence similarity from zero (blue) to 100% (red). Gray squares indicate no proteins that satisfied the selection criteria.

**Figure 3 ijms-24-08043-f003:**
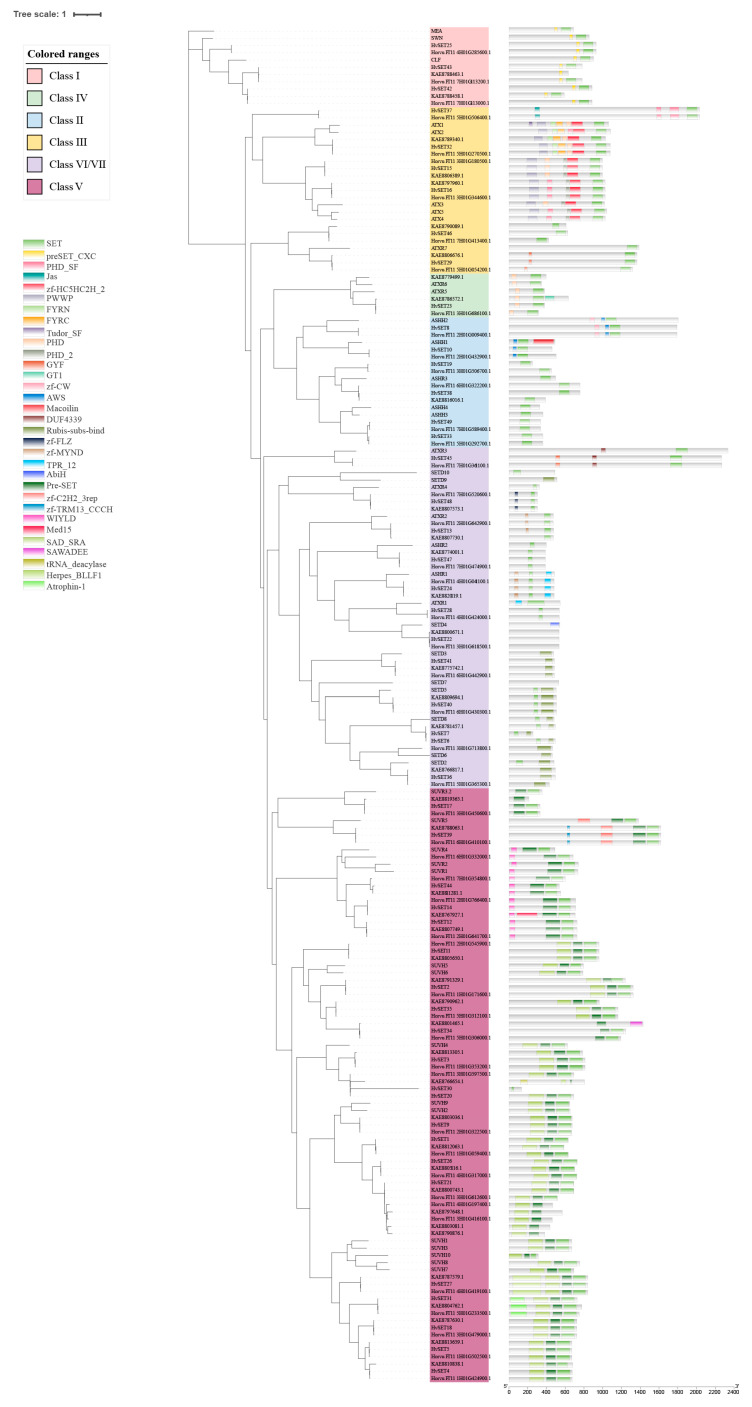
Phylogenetic and conserved domain analysis of SDG.

**Figure 4 ijms-24-08043-f004:**
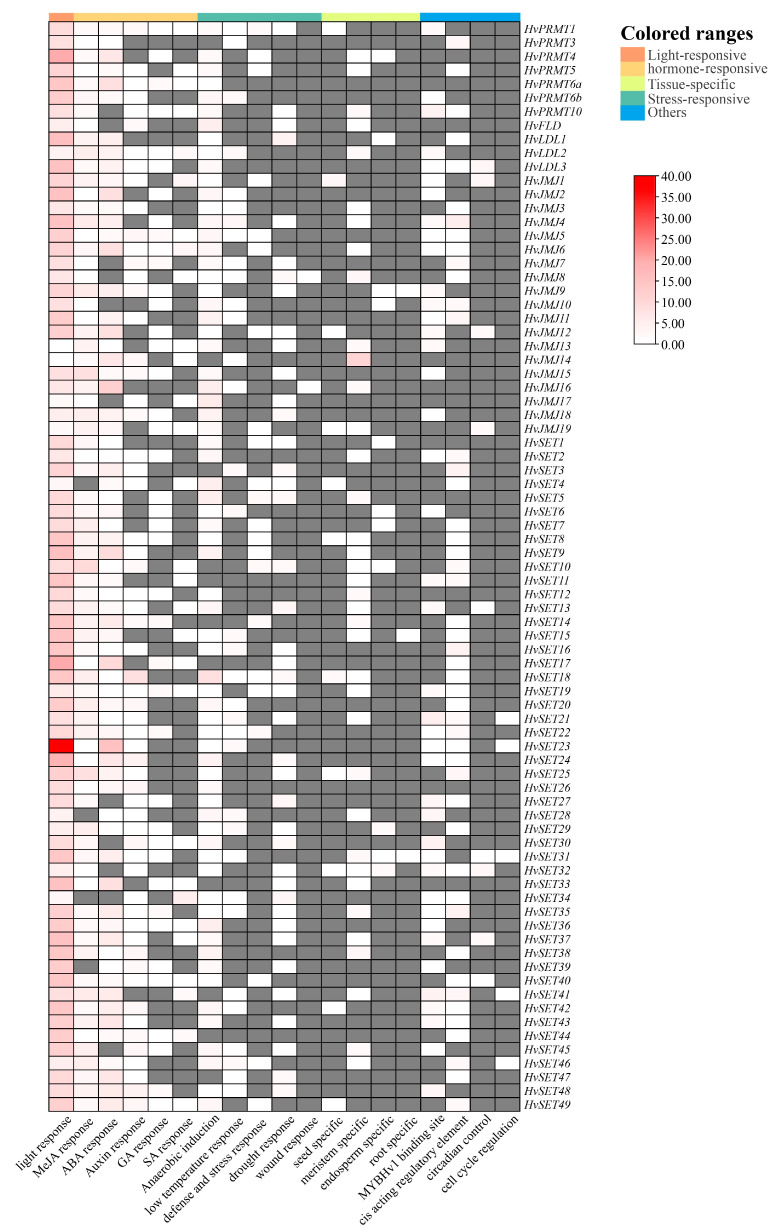
Analysis of promoter cis-acting elements of *HvHMTs* and *HvHDMs*. Gray squares indicates that the number of *cis*-acting elements in the gene is zero.

**Figure 5 ijms-24-08043-f005:**
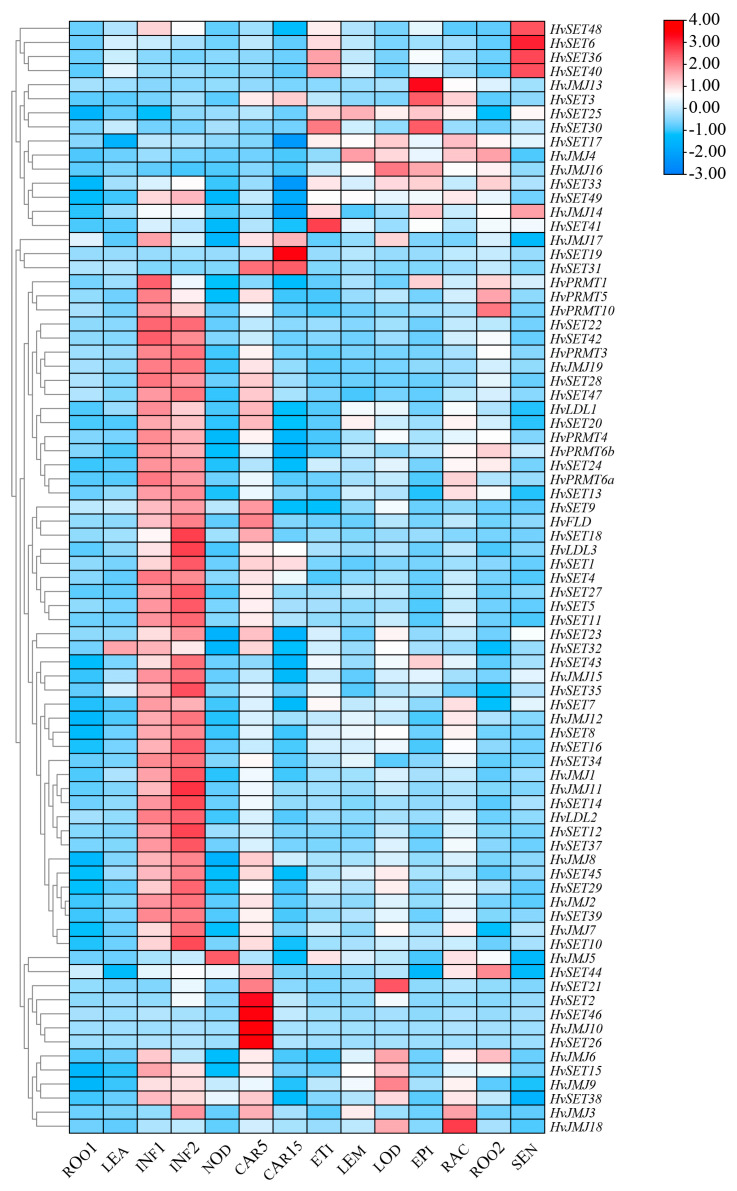
The spatiotemporal expression profile of *HvHMTs* and *HvHDMs* in different tissues or stages of barley. CAR15: bracts removed grains at 15 DPA; CAR5: bracts removed grains at 5 DPA; EPI: epidermis at 4 weeks old; ETI: etiolated from 10-day-old seedling; INF1: young inflorescences at 5 mm; INF2: young inflorescences at 1–1.5 cm; LEA: shoot at 10 cm from the seedlings; LEM: lemma at 6 weeks after anthesis; LOD: lodicule at 6 weeks after anthesis; NOD: developing tillers at six-leaf stage; RAC: rachis at 5 weeks after anthesis; ROO2: root from 4-week-old seedlings; ROO1: roots from the seedlings at 10 cm shoot stage; SEN: senescing leaf.

**Figure 6 ijms-24-08043-f006:**
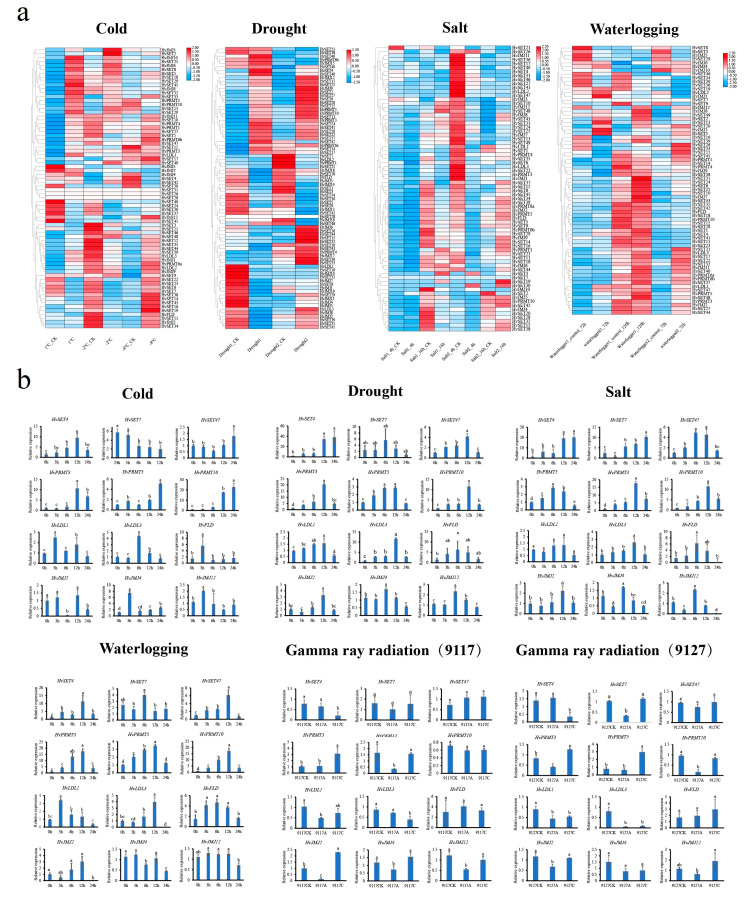
(**a**) RNA-seq: Cold, the expression patterns of *HvHMTs* and *HvHDMs* at 1 °C, −4 °C, and −8 °C. Drought, the expression patterns of *HvHMTs* and *HvHDMs* in sensitive varieties (Drought 1) and drought-tolerant varieties (Drought 2) under drought stress. Salt, the expression patterns of *HvHMTs* and *HvHDMs* in salt-sensitive varieties (Salt 1) and salt-tolerant varieties (Salt 2) under salt treatment for 4 h and 16 h. Waterlogging, the expression patterns of *HvHMTs* and *HvHDMs* of waterlogging tolerant varieties (Waterlogging 1) and medium waterlogging tolerant varieties (Waterlogging 2) under flooding. (**b**) RT-PCR: Expression of barley at 0, 3, 6, 12, and 24 h under low temperature, drought, salt, and flooding stress, and the expression of sensitive varieties (9127) and tolerant varieties (9117) under 50 Gy and 200 Gy intensity of gamma ray irradiation. Error bars represent ± SD from three biological repeats. Lowercase letters above bars indicate a significant difference (*p* < 0.05, LSD) among the treatments.

**Table 1 ijms-24-08043-t001:** Number of HMT and HDM family members in different barley germplasms.

Germplasm	Number of Family Members			
	HMT		HDM	
	SDG	PRMT	JMJ	HDMA
Akashinriki	51	7	18	4
B1K-04-12	50	7	18	4
Barke	52	7	18	4
Golden_Promise	51	7	17	4
Hockett	53	7	18	4
HOR_10350	51	7	18	4
HOR_13821	51	7	18	4
HOR_13942	51	7	18	4
HOR_21599	51	7	18	4
HOR_3081	50	7	18	4
HOR_3365	50	7	18	4
HOR_7552	52	7	18	4
HOR_8148	52	7	18	4
HOR_9043	51	7	18	4
Igri	51	7	18	4
Morex	49	7	19	4
OUN333	53	7	17	4
RGT_Planet	51	7	18	4
ZDM01467	50	7	18	4
ZDM02064	52	7	18	4
Lasa Goumang	42	7	17	4

## Data Availability

Not applicable.

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
