# Peer review of "Molecular Evolution of Histone Methylation Modification Families in the Plant Kingdom and Their Genome-Wide Analysis in Barley"

_ijms, 2023, doi:10.3390/ijms24098043_

Round 1
Reviewer 1 Report
Thank you for the opportunity to review the article entitled “Molecular evolution of histone methylation modification families and their genome-wide analysis in barley” which deals with a relevant and important topic.
I appreciate the experimental work in the paper and I feel that this research throws new light that deserves publication in a journal devoted to the latest finding in biochemistry, molecular and cell biology, molecular biophysics, molecular medicine, and all aspects of molecular research in chemistry.
The article made a positive impression on me. It is clear that the authors have done a great job. I miss the review of the theory. International works on this issue should be studied more. Also strengthen the link with the circular economy to provide a rich resource for future biological research and genomics-assisted breeding. Based on these comments, the article can be accepted for publication.
Reviewer 2 Report
The manuscript entitled “Molecular evolution of histone methylation modification families and their genome-wide analysis in barley” characterized the number of members, genetic similarity, and phylogeny of histone methyltransferases and histone demethylases families in barley using the barley pangenome and the Tibetan Lasa Goumang genome. The work is of importance since the results provide information of the regulatory role of HMT and HDM in barley growth and development and stress response and also provide information on the origin and evolutionary history of plant HvHMTs and HvHDMs.
I recommend the manuscript to be considered for publication after revision.
My concerns are:
The barley needs to be introduced properly. A more detailed description (some characteristics) in the introduction would be desirable.
L 4: Please use coma after each author
Please specify in the text the abbreviation for barley (Hv -Hordeum vulgare)
L 70-71: please provide de reference.
L 114-116: The sentence must be revised.
Figure S1: Please describe the full name for each abbreviation used.
L 122-126; 134-1236; 149-153; 162-164: These sentences should be placed in the Discussion section.
L 168, 170: ”the ” should not be italic.
L 176-182: Use the same tense to describe the results. Please be consistent throughout the manuscript.
Figure 2: The resolution of the figure is not high enough. Please provide a figure with better resolution, If possible. The same situation is also for figures 3 and 5.
L 198-194: Please revise the sentences. Probably the first sentence must be combined with the second, and the third does not make sense!
L243-246: “Hormone-sensitive elements in response to auxin, ABA, GA, MeJA and SA. Tissue-specific elements, including in the endosperm, meristems, roots, seeds and palisade mesophyll cells. Multiple abiotic and biotic stress-related elements, including in response to drought, anaerobic, hypothermia, injury, defense and stress.” Please revise the sentences. If the authors want to list a series of elements, use colons or another approach so that the phrases make sense.
Section 2.5. Analysis of cis-acting elements of HvHMTs and HvHDMs: the results needs to be presented properly.
L 252-268: The results are presented either to past or present. Please be consistent throughout the manuscript.
L 437-451: must be deleted.
L 506: There should be a protected space between “4” and “oC”, so that they are not separated by a new line.
L511-512: The sentence must be revised.
L517: Quantitative real-time PCR parameters must be specified.
Reviewer 3 Report
The overall manuscript is well written but I have some questions before making a final decision.
- What is the molecular evolution of histone methylation modification families in barley?
- How do the histone methylation modification families differ in their genome-wide distribution in barley?
- What are the functional implications of the genome-wide distribution of histone methylation modifications in barley?
- What is the relationship between histone methylation modifications and gene expression in barley?
- Are there any patterns in the histone methylation modification landscape that are conserved across different plant species?
- How do environmental factors influence the histone methylation modification landscape in barley?
Round 2
Reviewer 3 Report
All concerns are addressed.